# Cold Plasma Treatment for Efficient Control over Algal Bloom Products in Surface Water

**Hee-Jun Kim [1], Gui-Sook Nam [2], Jung-Seok Jang [2], Chan-Hee Won [1],* and Hyun-Woo Kim [1],*** 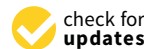

[1] Department of Environmental Engineering, Division of Civil, Environmental, Mineral Resource and Energy Engineering, Soil Environment Research Center, Chonbuk National University, 567 Baekje-daero, Deokjin-gu, Jeonju 54896, Korea

[2] Rural Research Institute, Korea Rural Community Corporation, 870 Haean-ro, Sangnok-gu, Ansan 15634, Korea

* Correspondence: chwon@jbnu.ac.kr (C.-H.W.); hyunwoo@jbnu.ac.kr (H.-W.K.); Tel.: +82-63-270-2446 (C.-H.W.); +82-63-270-2444 (H.-W.K.); Fax: +82-63-270-2449 (H.-W.K.)

**Abstract:** Algal bloom significantly alters the physicochemical properties of water due to drastic pH change, dissolved oxygen depletion/super-saturation, and toxicity, which lead to ecosystem destruction. To prevent this, this study evaluated the reduction performance of algal biomass by applying a non-thermal or cold plasma process. We used chlorophyll-a (chl-a), suspended solids (SS), and turbidity as indicators of the biomass. Results demonstrated that their removal efficiencies were in the ranges 88–98%, 70%–90%, and 53%–91%, respectively. Field emission scanning electron microscopy indicated how the cell wall of microalgae was destroyed by cold plasma. Also, the removal kinetics of cold plasma confirmed the enhanced removal rate constants. The estimated required times for 99% removal were 0.4–1.2 d (chl-a), 1.3–3.4 d (SS), and 1.6–6.2 d (turbidity), respectively. Overall, cold plasma could be a useful option to effectively treat pollution associated with algal bloom in surface water.

**Keywords:** algal bloom; chlorophyll-a; suspended solids; turbidity; advanced oxidation

## 1. Introduction

Microalgal bloom associated with the rapid growth of numerous algae and cyanobacteria in surface water has been frequently reported in receiving water worldwide [1]. In addition to the insufficient management of eutrophication by incoming non-point sources [2,3], climate change and global warming are also major causes of the significant blooms [1,4,5]. If the bloom occurs at around the water intake, it leads to serious drinking water quality problems [6], and thus securing the high quality of water resources from the bloom is currently a very important issue [7]. The bloom changes the physicochemical properties and microbial communities of surface water [8] due to drastic pH change and dissolved oxygen depletion/super-saturation, leading to fish deaths, bad taste, odor-causing compounds, and toxins [9,10].

These problematic issues made by microalgal biomass have become the main removal targets in surface water [11,12]. Therefore, newly suggested methods have been tested, such as modified local soil technology [3], advanced oxidation processes (AOPs) [13], and so on. In most cases, the amount of biomass has been quantified by either suspended solids (SS) [14], turbidity, or chlorophyll-a (chl-a) [15], and thus the control over the major indicator pollutants is crucial for water resources protection and security.

Various AOPs have been applied to remove microalgal biomass, but UV treatment has a limitation when the turbidity of the raw water is high [16], and $O_3$ has difficulties in controlling the pH [17].

In the case of Fenton oxidation, excess sludge generation is problematic when the organic content is too high [18]. For ultrasonication and microwave treatment, cost can be a limiting factor despite the strong physicochemical oxidizing power [19].

Among various options, the non-thermal or cold plasma process has been known to overcome the disadvantages of conventional AOP. Cold plasma is known to cause a variety of oxidative and chemical species, such as radicals (e.g., H·, O·, and OH· ) and molecules (e.g., $H_2O_2$ and $O_3$), shock waves, ultraviolet, and electrohydraulic cavitation [13,20–23]. In addition, the cold plasma process is considered to be a process combining the advantages of other AOPs, and has relatively few constraints on temperature and pressure [24–26]. However, the cold plasma process for the removal of microalgae has rarely been reported.

This study therefore attempted an air-assisted cold plasma process as a method to treat indicators of microalgal biomass content. Experiments were designed to confirm the efficient elimination of algae by the cold plasma. Their specific objectives were: (1) to verify how the cold plasma effectively removes chl-a, SS, and turbidity at various initial concentrations; (2) to investigate the kinetic rate constants of the cold plasma process; (3) to find cause and effect of microalgal cell change in the cold plasma treatment; and (4) to estimate time for 99% removal based on the kinetics.

## 2. Materials and Methods

### 2.1. Experiment Set-Up

The water samples were taken from a local reservoir that suffered from algal bloom in Korea. The characteristics of the sample are shown in Table 1.

**Table 1.** Physical and chemical characteristic of sampled raw surface water. $COD_{cr}$: chemical oxygen demand; TOC: total organic carbon; SS: suspended solids; TN: total nitrogen; Chl-a: chlorophyll-a.

| Item | Average |
| --- | --- |
| $COD_{cr}$ (g/L) | 4.1 |
| TOC (g/L) | 0.4 |
| SS (g/L) | 1.2 |
| TN (g/L) | 0.3 |
| Turbidity (NTU) | 3600 |
| Chl-a ($g/m^3$) | 10.8 |

A lab-scale cold plasma process equipped with the direct aeration of carrier gas and air was applied to the sample. The air-assisted cold plasma was provided by Groon co. Ltd., in Jeonju city, Korea. The glow-discharge-based plasma generator (10 mA and 2.2 W) produced plasma in the passing air. The aeration pump (1.5 A and 330 W, WELCH, Model No. 2546 C-10) was operated with the air flowrate of 5 L/min, which was controlled by a flow meter (Dwyer, RMA-22-SSV, USA). After the air passed through a plasma-generating apparatus, reactive chemicals in the plasma state were directly contacted with the target pollutants in the water sample. A schematic diagram of the device is shown in Figure 1.

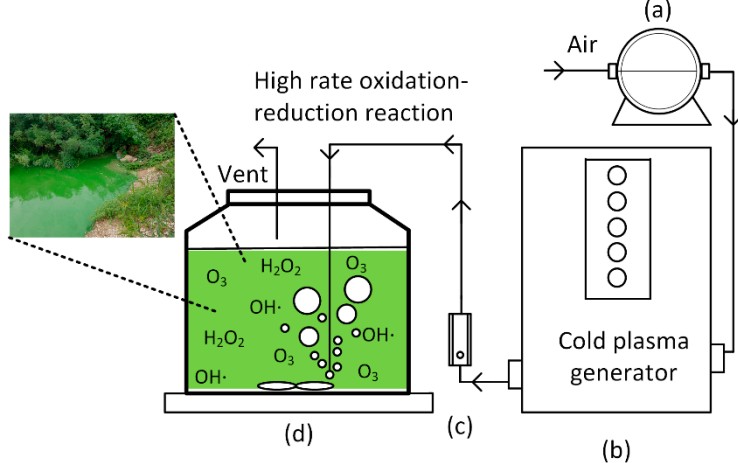

**Figure 1.** Lab-scale set-up of the air-assisted cold plasma process for batch experiments at an air flowrate of 5 L/min: (**a**) aerator, (**b**) cold plasma generator, (**c**) flow meter, and (**d**) magnetic stirring and reactor.

To investigate the removal efficiencies of indicator parameters, we diluted the original sample for five different runs, as summarized in Table 2. Under each initial concentration, cold plasma treatments were conducted for 24 h, and the samples were collected according to the experimental design for analyses.

**Table 2.** Mixing ratio of raw surface water to dilution (distilled water).

| Run | Dilution Ratio [a] (times) |
| --- | --- |
| 1 | 8.0 |
| 2 | 4.0 |
| 3 | 3.0 |
| 4 | 1.3 |
| 5 | 1.0 |

[a] Total water volume after dilution/raw water volume.

## 2.2. Experimental Measurements

Using spectrophotometry, chl-a concentrations were analyzed by a method based on acetone extraction [27]. To quantify the total mass of microalgae, SS was determined by adapting 2540 D of the standard method for the examination of water and wastewater [28]. Turbidity was determined by a turbidity meter (Orion AQ4500, Thermo Scientific, Singapore), which recorded nephelometric turbidity units (NTUs).

The microalgal cell surface was observed using a field emission scanning electron microscope (FE-SEM). After centrifugation, the cells were resuspended by inserting a fixing solution containing 2% paraformaldehyde and 2% glutaraldehyde into 0.05 M cacodylate buffer (pH 7.2). We then kept the sample at room temperature overnight. Thereafter, cells were washed three times (10 min each) with 0.05 M sodium cacodylate buffer (pH 7.2). Then, cells were immersed in 1% osmium tetraoxide in 0.05 M sodium cacodylate buffer, pH 7.2, for 1.5 h at 4 °C. Again, after two times of washing with distilled water, stepwise dehydration in ascending ethanol content (30%, 40%, 50%, 70%, 80%, 90%, and 100%) was conducted. Overall, the pretreatment process took approximately 80–90 min to complete. The pretreatment was followed by a drying processes of: (1) two steps of chemical drying were conducted with 100% hexamethyldisilazane (HMDS) for 15 min each, and then (2) two rounds of critical point drying were conducted with 100% isoamyl acetate for 15 min. Afterwards, a gold film coating made the cell surface observable by FE-SEM equipment (SUPRA 40 VP, Carl Zeiss Oberkochen, Germany).

*2.3. Statistical Analyses and Regressions*

To determine the removal rate constant, k (d$^{-1}$), the computer software SigmaPlot (Systat Software, Inc., San Jose, CA, USA) was used for regression based on first-order exponential decay assumptions of chl-a, SS, and turbidity over the cold plasma operation time. To estimate the response time for 99% removal of chl-a, SS, and turbidity, we replaced the concentration of the first-order exponential decay model as 0.01c0 with the obtained k (d$^{-1}$), where c0 is the initial concentration of either chl-a, SS, or turbidity.

## 3. Results and Discussion

*3.1. Degradation of Major Microalgal Indicators by Cold Plasma Application*

The experiment was designed to verify the degradation performance of microalgal indicators by cold plasma. Experimental results were obtained based on the experimental design. Figure 2 shows the degradation trends of chl-a, SS, and turbidity according to the cold plasma contact time.

As shown in Figure 2a, the initial chl-a of Run 5 was 10.8 g/m$^3$, which decreased to 1.2 g/m$^3$ after 24 h, having an 88.8% removal efficiency. The chl-a of Run 4 started at 7.9 g/m$^3$, and the final concentration after 24 h was merely 0.5 g/m$^3$ (94.3% removal efficiency). Run 3 showed the highest removal efficiency of 98.8%, equivalent to a decrease from 6.4 g/m$^3$ to 0.08 g/m$^3$. In the case of Run 2 and Run 1, initial chl-a (3.1 g/m$^3$ and 1.0 g/m$^3$) declined to 0.07 g/m$^3$ (97.7% removal) and 0.05 g/m$^3$ (95.1% removal), respectively.

Figure 2b illustrates the dynamics of SS in the reactor. It was revealed that the cold plasma actively degraded suspended solids in the reactor, though Run 5, with the highest initial concentration, presented an initial fluctuation. The SS of Run 5 was measured to be 1.28 g/L, and after 24 h treatment the SS value was 0.29 g/L, showing a 77.3% removal efficiency. The SS values of Run 4 and Run 3 were measured as 1.01 g/L and 0.58 g/L, respectively. The final data were 0.15 g/L and 0.08 g/L. The removal efficiencies of the two runs were only 85.0% and 85.3%, respectively. Run 2 showed the highest removal efficiency of 90.5% (0.42 g/L to 0.04 g/L). The SS value of Run 1 was 0.1 g/L, and it decreased to 0.03 g/L, recording a 70.0% removal efficiency.

Figure 2c presents the turbidity according to the cold plasma contact time. The initial turbidity of Run 5 was 3267 NTU, and the final was recorded as 1519 NTU, showing a 53.5% removal. The initial turbidity of Run 4 (1800 NTU) reduced to 460 NTU (74.7%) after 24 h. Meanwhile, Run 3 showed the highest removal efficiency of 91.1% (1275 NTU to 113 NTU). The lower initial turbidities of Run 2 (346 NTU) and Run 1 (108 NTU) did not lead to higher removal efficiency. The final turbidities were 84.6 NTU (75.5%) and 41.3 NTU (61.8%), respectively.

Previous studies have shown that the AOP process was effective in chl-a removal when the initial concentration was ~160 mg/m$^3$ [29,30], and was also shown to be effective in SS removal [25,31–33] and turbidity removal [14,34–36].

This study found that cold plasma could obtain chl-a removal efficiencies of over 94% when the initial concentration was higher than 1.4 g/m$^3$ (Run 1). This result implies that cold plasma is very effective in chl-a reduction, even at extremely high initial chl-a (~10 g/m$^3$). The obtained removal efficiencies indicate that the reactivity of chemical species from cold plasma seems to be very strong, since this study demonstrated little difference in efficacy for the tested range of initial chl-a.

Results also showed SS removal greater than 70% for all the experiments, indicating the efficacy of the cold plasma process, though a slight increase of SS (1.08 to 1.24 g/L) at 3 h was observable for higher initial SS (Run 5). This is possibly because algal cell disintegration, extracellular by-products separation, or intercellular materials release may increase or maintain the initial SS due to active simultaneous reactions.

The strong oxidizing power of the cold plasma process similarly confirmed the reduction of turbidity, though the treatment efficiencies were lower than those of chl-a and SS. In the case of Run 5, the turbidity increased to 4300 NTU during first 3 h and then started to decrease. This increase may

be associated with the release of inner cell materials in the microalgae decomposition process, which is consistent with the literature [29]. These results indicate that the turbidity is not a good indicator for biomass removal assessment compared to chl-a and SS, though the cold plasma process could effectively oxidize turbidity, eventually obtaining material degradations of up to 91.1%.

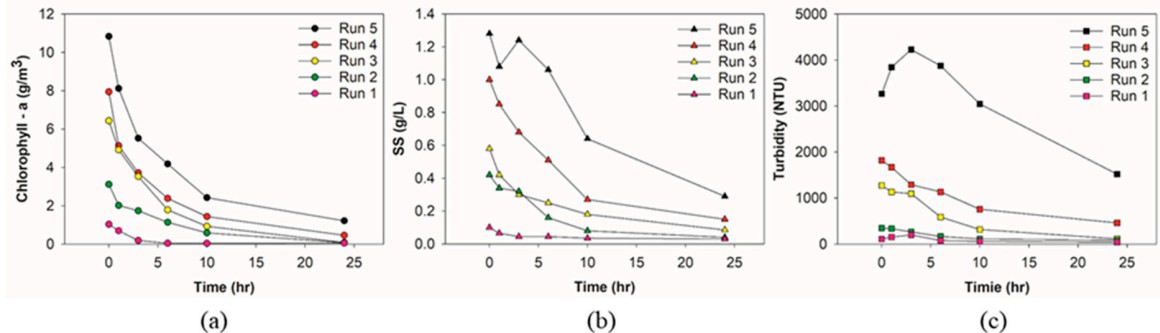

**Figure 2.** Concentration value by item according to each run and contact time: (**a**) chl-a; (**b**) SS; (**c**) turbidity.

### 3.2. Enhanced Degradation Kinetics in Cold Plasma Process

The removal rate constant ($k_{chl-a}$) of Run 1 with a relatively low initial concentration (1 g/m$^3$) was 3.75 d$^{-1}$ (Table 3). When the chl-a concentration was over 10.8 g/m$^3$ (Run 5) it was 11.84 d$^{-1}$—approximately three times slower due to the increased concentration of chl-a. However, Runs 3, 4, and 5 showed slightly lower or similar removal rates than Run 2, which indicates that the removal rate could be maintained despite the increase of biomass being removed (Figure 3a). Therefore, it is considered that a high removal rate can be obtained at a low concentration, and a large amount of removal can be obtained at a high concentration. This large variation occurs possibly due to the difference in the initial amount of chl-a, but all cases showed higher values than that (3.55 d$^{-1}$, $R^2$ = 0.97) of UV-C degradation [37]. Based on the regression, removal rates (k) of chl-a were estimated.

The removal rate constant of SS showed a different pattern to chl-a (Figure 3a,b). The highest removal rate of Run 4 was 3.38 d$^{-1}$. Depending on concentration, the removal rate constant ranged between 1.35 d$^{-1}$ and 3.38 d$^{-1}$ (Table 3). Especially, when the concentration of SS was approximately 100 mg/L (Run 1), a very low removal rate of 1.34 d$^{-1}$ was recorded. It was judged that the removal rate was the lowest because of a low initial amount of SS. Therefore, the removal rate might be lowered due to the overload of particulate matter in this SS removal by cold plasma.

**Table 3.** The removal rate constant, k, was regressed by using the data obtained from each experimental condition of chl-a, SS, and turbidity.

| Run | Chl-a | | | SS | | | Turbidity | | |
|---|---|---|---|---|---|---|---|---|---|
| | Experimental Removal Efficiency (%) | Regressed Removal Rate, k (d$^{-1}$) | $R^2$ | Experimental Removal Efficiency (%) | Regressed Removal Rate, k (d$^{-1}$) | $R^2$ | Experimental Removal Efficiency (%) | Regressed Removal Rate, k (d$^{-1}$) | $R^2$ |
| 1 | 95.1 | 11.84 | 0.96 | 70.0 | 1.35 | 0.90 | 61.8 | 0.73 | 0.71 |
| 2 | 97.7 | 3.92 | 0.96 | 90.5 | 2.69 | 0.98 | 75.5 | 1.74 | 0.96 |
| 3 | 98.8 | 4.80 | 0.99 | 85.3 | 2.89 | 0.92 | 91.1 | 2.85 | 0.96 |
| 4 | 94.3 | 4.75 | 0.96 | 85.0 | 3.38 | 0.96 | 74.7 | 2.39 | 0.94 |
| 5 | 88.8 | 3.75 | 0.99 | 77.3 | 1.92 | 0.66 | 53.5 | 1.44 | 0.52 |

The removal rate constants of turbidity presented a slightly similar trend to that of SS (Figure 3b,c). Estimated removal rate constants of Runs 3 and 4 presented little difference, at 2.85 d$^{-1}$ and 2.39 d$^{-1}$, respectively (Table 3). Only Run 5 with the highest turbidity and Run 1 with the lowest turbidity showed significantly reduced removal rates (1.44 d$^{-1}$ and 0.73 d$^{-1}$).

This is a distinctively different removal trend from those of chl-a, which showed the highest removal rate in Run 1 (Figure 3a). This could be attributed to the nature of turbidity measurement, due to a wide range of particles causing incident light scattering. The residual cell wall (Figure 4c) and inner cell materials (Figure 2b,c) due to cold plasma application must have different properties in their shape, color, and reflectivity, leading to an unreliable determination for very high and low turbidity (Run 5 and 1).

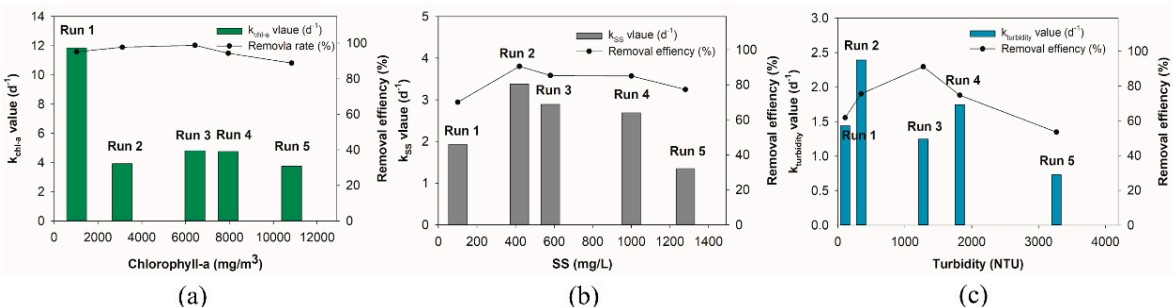

(a)                                         (b)                                         (c)

**Figure 3.** Regressed degradation kinetics rates according to the initial concentration and the removal efficiency at the end of each run: (**a**) chl-a ($k_{chl-a}$); (**b**) SS ($k_{SS}$); (**c**) turbidity ($k_{turbidity}$).

### 3.3. Morphological Change of Microalgal Cell Surface by Cold Plasma

To verify the removal mechanism of chl-a, FE-SEM pictures of the cell surface was taken before and after the cold plasma treatment. Figure 4 illustrates the representative changes found on the cell surface. The pictures clearly show that the cold plasma caused substantial destruction of the microalgal cell wall. Especially, Figure 4b,c demonstrates the complete breakage and shrinkage of the cell surface, which implies the devastating decomposition of the inner cell materials including chl-a, though a residual skin-like cell wall still remained.

This indicates that the oxidation and reduction of reactive chemicals produced by the cold plasma can lead to bactericidal or bacteriolytic inactivation of the cells. This is consistent with previous results [26,30] and may suggest a stronger effect compared to previous literature. Figure 4c demonstrates that the changes of the cell surface caused severe damage to the cell surface due to continuous exposure to reactive chemical species. These results support the efficient inactivation of microorganisms in the cold plasma treatment.

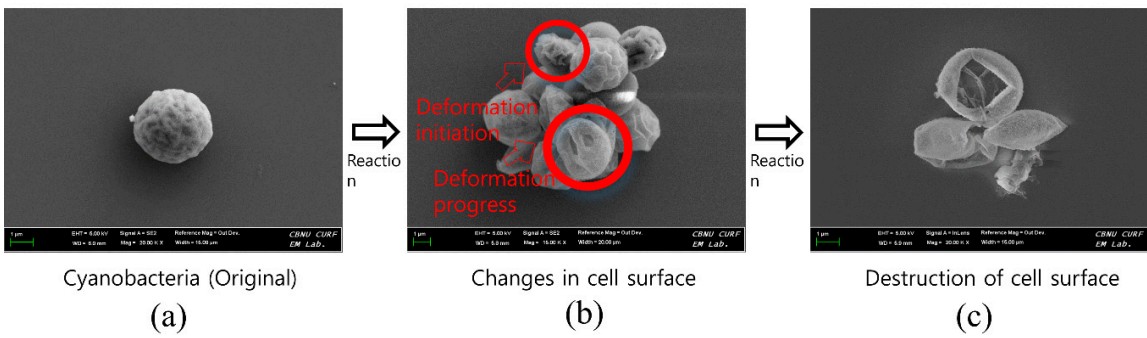

Cyanobacteria (Original)            Changes in cell surface            Destruction of cell surface

(a)                                         (b)                                         (c)

**Figure 4.** SEM images showing morphological destruction of the microalgal cell by reactive chemicals from the cold plasma. (**a**) Original single cell before cold plasma application; (**b**) shrunken cell surface after cold plasma application; and (**c**) torn-off cell surface at the end of the cold plasma treatment.

### 3.4. Necessity of Enough Contact Time for Bloom Control

Figure 4c shows that cell wall debris of microalgae remained, constituting the remaining SS. This explains the low treatment efficiency of SS compared to the case of chl-a. It is evident that removal rates of chl-a for Runs 3, 4, and 5 were highly maintained, while those of SS were relatively low. So,

it is judged that the removal was still effective even though the SS removal efficiency was lower than expected because of the residual SS. The cold plasma process was limited in its capability of complete SS reduction within a short contact time (24 h). Thus, the 99% response times for SS were much longer (1.3–3.4 d) than those for chl-a (Table 4).

**Table 4.** The removal rate k calculated using the data obtained from each experimental condition of chl-a, SS, and turbidity.

| | Chl-a | SS | Turbidity |
|---|---|---|---|
| Run | Estimated 99% Response Time [a] (d) | Estimated 99% Response Time [a] (d) | Estimated 99% Response Time [a] (d) |
| 1 | 0.4 | 3.4 | 6.2 |
| 2 | 1.2 | 1.7 | 2.6 |
| 3 | 0.9 | 1.6 | 1.6 |
| 4 | 0.9 | 1.3 | 1.9 |
| 5 | 1.2 | 2.4 | 3.1 |

[a] 99% response time means that the concentration was lowered to 99% of its initial value.

The cloudy transparency of the cold-plasma-treated water (Figure S1 in Supporting Information) also supports the potential interference due to the remaining fine particles or dissolved solids from the degradation of biomass-related compounds. For these reasons, the 99% response time was extended significantly to ~6.2 d (Table 4). For Run 1, the initial turbidity was only 108 NTU, and a decrease to about 1 NTU would be required for a 99% removal efficiency. Therefore, more contact time was required to obtain such a low turbidity.

## 4. Conclusions

This study revealed that the strong oxidizing power from reactive chemicals produced by cold plasma led to excellent microalgal removal. Our experimental results based on chl-a, SS, and turbidity determinations as biomass indicators demonstrated substantially high removal efficiency (over 90.5%). Removal rates indicated that SS and turbidity tend to differ from chl-a due to the particulate matter produced by the degradation of large amounts of microalgal cell. FE-SEM analysis indicated that the removal mechanisms of chl-a and SS were due to severe deformation of the microalgal cell surface. The minimum time required to achieve a 99% removal efficiency for each indicator was estimated at approximately 1.6 d. Overall, this study verified the advanced performance of cold plasma from the viewpoints of chl-a, SS, and turbidity removal, and provides their kinetics. It may provide a sustainable approach to control a wide range of microalgal blooms without concerns about microalgal biomass harvesting.

**Supplementary Materials:** The following are available online at http://www.mdpi.com/2073-4441/11/7/1513/s1, Figure S1: Effect of cold plasma: (A) before treatment; and (B) after treatment (24 h).

**Author Contributions:** H.-J.K., J.-S.J., and H.-W.K. conceived and designed the experiments; H.-J.K. and G.-S.N. performed all experiments; H.-J.K., C.-H.W., and H.-W.K. analyzed the data and wrote the paper; C.-H.W. and H.-W.K. took leadership of the project and the publication of the paper; All authors read and approved the submitted manuscript.

**Funding:** This research was funded by the Korean Ministry of Environment (MOE) as part of the 'Knowledge-based environmental service (Waste to energy recycling) Human resource development Project'. This research was also supported by research funds of Chonbuk National University in 2019.

**Acknowledgments:** We thank the Center for University Research Facility (CURF) at Chonbuk National University for helpful support and comments in FE-SEM analyses.

**Conflicts of Interest:** The authors declare no conflicts of interest.

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
