# Peer review of "Cold Plasma Treatment for Efficient Control over Algal Bloom Products in Surface Water"

_water, doi:10.3390/w11071513_

Round 1
Reviewer 1 Report
The authors present and evaluation of the use of cold plasma as a water purification process. The paper is well-written, appropriately referenced with useful tables and figures, and generally acceptable for publication. However, this reviewer would strongly recommend that the authors continue their research and include an examination of algal intracellular/extracellular products as a possible consequence of plasma treatment. One of the most important aspects of water purification is the removal of algal toxins, especially where the source water is enriched, with heavy algal blooms. Lysing the cells can release the algal toxins into the product water. These toxins may not show up as turbidity, but could severely impact water users. That said, the paper represents a good first step in the evaluation of the use of cold plasma and is worthy of publication. One final point, deoxygenation is not the only impact of algal blooms that involves dissolved oxygen in the water column: oxygen super-saturation can also occur during daylight hours, with adverse effects on fish whose gill structures are disrupted by too much oxygen.
Author Response
POINT-BY-POINT RESPONSE TO EACH REVIEWER’S COMMENTS
Authors: Hee-Jun Kim, Gui-Sook Nam, Jung-Seok Jang, Chan-Hee Won, and Hyun-Woo Kim
Manuscript ID: Water-542057
Manuscript Title: Cold plasma treatment for efficient control over algal bloom products in surface water
We appreciate the many helpful comments from the reviewer and revised the manuscript following the reviewer’s comments as we indicate below. Except for trivial changes, we highlight our revisions using red color in the manuscript and in this response memo, as suggested by the editor.
Reviewer #1:
The authors present and evaluation of the use of cold plasma as a water purification process. The paper is well-written, appropriately referenced with useful tables and figures, and generally acceptable for publication. However, this reviewer would strongly recommend that the authors continue their research and include an examination of algal intracellular/extracellular products as a possible consequence of plasma treatment. One of the most important aspects of water purification is the removal of algal toxins, especially where the source water is enriched, with heavy algal blooms. Lysing the cells can release the algal toxins into the product water. These toxins may not show up as turbidity, but could severely impact water users. That said, the paper represents a good first step in the evaluation of the use of cold plasma and is worthy of publication.
Response: We would like to thank thoughtful comments by the reviewers, which enabled us to make a minor revision to improve our manuscript. Our team has carried out preliminary tests to quantify the release of algal toxins and to trace the fate of the intracellular products focusing on toxicity change. These were extensive studies on their own, and we are planning to report the results separately, as suggested by reviewer 1.
One final point, deoxygenation is not the only impact of algal blooms that involves dissolved oxygen in the water column: oxygen super-saturation can also occur during daylight hours, with adverse effects on fish whose gill structures are disrupted by too much oxygen.
Response: As noted by the reviewer 2, we admit that the supersaturation of DO can also be an indicator of algal bloom. Thus, we address Reviewer 1’s concerns on this issue in our responses by augmenting the expression “depletion” into “depletion/super-saturation” in the ‘Abstract’ and ‘Introduction’ to make clearer our meaning, as we show below.
(Abstract line 2) “…, dissolved oxygen depletion/super-saturation,…”
(Introduction line 9) “… and dissolved oxygen depletion/super-saturation leading…”

Reviewer 2 Report
Recommendation: This paper is publishable subject to minor revisions noted. Further review is not needed.
Comments:
The paper reports the modern approach to water purification ( microalgae in raw water). The work provides insight into Cold Plasma treatment of . The writers give a deep analysis and discuss possible mechanisms of purification of water by plasma process.
Authors provide the analysis data of chlorophyll-a , suspended solids , and turbidity before and after treatment, they also investigate kinetic rate constants of the cold plasma process.
Should be noted brilliant work on the provision of high quality micrographs of algae cells ,shows crucial role of plasma process on membrane morphology.
Remarks:
Suggest using complete name of company providing the plasma unit and the full city name (line 63)
I recommend the acceptance of the paper after minor correction.
Author Response
POINT-BY-POINT RESPONSE TO EACH REVIEWER’S COMMENTS
Authors: Hee-Jun Kim, Gui-Sook Nam, Jung-Seok Jang, Chan-Hee Won, and Hyun-Woo Kim
Manuscript ID: Water-542057
Manuscript Title: Cold plasma treatment for efficient control over algal bloom products in surface water
We appreciate the many helpful comments from the reviewer and revised the manuscript following the reviewer’s comments as we indicate below. Except for trivial changes, we highlight our revisions using red color in the manuscript and in this response memo, as suggested by the editor.
Reviewer #2:
Recommendation: This paper is publishable subject to minor revisions noted. Further review is not needed.
Comments: The paper reports the modern approach to water purification (microalgae in raw water). The work provides insight into Cold Plasma treatment of. The writers give a deep analysis and discuss possible mechanisms of purification of water by plasma process. Authors provide the analysis data of chlorophyll-a, suspended solids, and turbidity before and after treatment, they also investigate kinetic rate constants of the cold plasma process. Should be noted brilliant work on the provision of high quality micrographs of algae cells, shows crucial role of plasma process on membrane morphology.
(Page 2, Line 41) Remarks: Suggest using complete name of company providing the plasma unit and the full city name (line 63)
Response: As suggested by the reviewer 2, we supply the requested information of the company and city name.
“The air-assisted cold plasma was provided by Groon co. ltd., in Jeonju city, Korea.”
